# Physician and Patient Dissatisfaction with Outpatient Pre-Screening Triage in Public Dental Hospitals: Scope and Strategies for Improvement

**DOI:** 10.3390/healthcare13141672

**Published:** 2025-07-11

**Authors:** Siwei Ma, Li Zhang, Wenzhi Du, Gaofeng Fang, Peng Zhang, Fangfang Xu, Xingke Hao, Xiaojing Fan, Ang Li

**Affiliations:** 1Key Laboratory of Shaanxi Province for Craniofacial Precision Medicine Research, College of Stomatology, Xi’an Jiaotong University, Xi’an 710004, China; masiwei@mail.xjtu.edu.cn (S.M.); duwenzhi6442@dentalxjtu.com (W.D.); fanggaofeng@xjtu.edu.cn (G.F.); 376686331@xjtu.edu.cn (P.Z.); xff@mail.xjtu.edu.cn (F.X.); haoxingke@dentalxjtu.com (X.H.); 2Department of Operations & Management, College of Stomatology, Xi’an Jiaotong University, Xi’an 710004, China; 3School of Public Policy and Administration, Xi’an Jiaotong University, Xi’an 710049, China; zhangli2023@stu.xjtu.edu.cn

**Keywords:** dissatisfaction, pre-screening triage, public hospital, physician and patient perspectives

## Abstract

**Objectives:** While pre-screening triage (PST) enhances healthcare efficiency in emergency and pediatric settings, its application in dental healthcare remains undervalued. This novel study implemented PST in dental services, identifying determinants of physician–patient dissatisfaction to optimize triage systems and promote dental health outcomes. **Methods:** A cross-sectional survey (July–September 2024) recruited 113 physicians and 206 patients via convenience sampling. Dissatisfaction levels were quantified using validated questionnaires and analyzed through *t*-tests, ANOVA, and regression models. **Results:** In total, 37.17% of physicians with prior PST experience demonstrated significantly higher dissatisfaction scores (37.67 ± 9.08 vs. 32.51 ± 10.08, *p* = 0.006). Multivariate analysis revealed that experienced physicians rated PST services 5.63 points higher than less experienced counterparts (95% CI: 0.75–10.51). Dental patients expressed dissatisfaction with nurse attitudes (β = 1.04, 95% CI: 0.07–2.01) and triage process inefficiencies. **Conclusions:** Key dissatisfaction drivers include a lack of physician PST exposure and nurse–patient interaction quality in dental settings. These findings advocate for the development of a specialized triage system to enhance clinical workflow efficiency and service effectiveness in dental healthcare.

## 1. Introduction

Pre-screening and triage (PST) is a critical mechanism for optimizing healthcare delivery in public hospitals, aiming to balance equitable resource allocation with clinical efficiency [1,2]. While traditional triage conducted by registration staff focuses on administrative tasks—such as department assignment and consultation numbering—this fragmented approach often fails to address systemic inefficiencies in the patient journey [3,4]. Patients may endure repeated assessments across departments, resulting in delays in critical care prioritization, or face mismatches between symptom urgency and specialist availability—all of which erode trust in the healthcare system and strain hospital-wide workflows [5,6]. In contrast, PST integrates systematic evaluation of medical history, symptom patterns, and clinical measurements to establish a severity-based prioritization, streamlining the patient pathway [1,2]. PST thereby addresses two interconnected gaps: patient-flow bottlenecks caused by siloed department-level triage and clinical decision-making delays due to incomplete symptom documentation [7,8]. By synchronizing prioritization with structured symptom documentation, PST not only reduces waiting times for high-risk cases but also enhances cross-departmental coordination, improving both individual patient outcomes and operational resilience.

In the realm of healthcare research, previous research into PST has predominantly focused on its utilization within specific clinical departments. Research on PST in emergency departments mainly focuses on how to shorten the waiting time to triage patients as soon as possible and improve the quality of emergency treatment [9,10]. Pediatrics is another area that has received considerable attention. Given the unique characteristics of pediatric patients, including their inability to fully communicate symptoms and the rapid progression of certain childhood diseases, PST plays a vital role in promptly assessing the severity of a child’s condition and reducing diagnostic delays [8]. Studies in infectious disease management validate PST’s efficacy in early pathogen exposure identification [8,11]. However, when it comes to the broader perspective of PST application at the overall hospital level, there has been a relative dearth of research. This gap in the literature highlights the need for more comprehensive studies that explore how PST can be effectively integrated across different departments and services within a hospital to optimize patient flow and improve overall healthcare delivery.

These department-specific optimizations inadvertently create systemic fragmentation. Fewer studies focused on the PST application at the overall hospital level. In this study, the Stomatological Hospital of Xi’an Jiaotong University took an innovative approach. We integrated PST into the first consultation of dental healthcare services and collected two-way dissatisfaction evaluations from physicians and patients through questionnaires after implementation. This study aims to identify strategies that could enhance the application and efficiency of PST to streamline healthcare services and improve the dental health of the Chinese population.

## 2. Methods

### 2.1. Data Source

Data were obtained from an anonymous cross-sectional online survey named “Evaluation Questionnaire for PST Services in Public Hospitals from the Perspectives of Physicians and Patients”, which was conducted between 1 July and 30 September 2024. The survey questionnaire was self-designed.

The design of the questionnaire was based on a literature review, where we systematically identified the factors related to doctors’ and patients’ dissatisfaction within hospitals, while also focusing on the impact factors brought about by the implementation of new policies [12,13,14,15]. On this basis, we developed two draft survey questionnaires from the perspectives of doctors and patients, focusing on key dimensions such as the triage process, doctor–patient communication, and patients’ lived experiences. These questionnaires were scored using a 5-point Likert scale.

Second, to ensure the scientific rigor, rationality, and practicality of the questionnaire, a panel of five senior scholars from multiple fields, including hospital management, health policy, and oral medicine, and ten healthcare professionals with extensive clinical experience, was organized. Leveraging their professional knowledge and practical experience, they conducted in-depth discussions and meticulous revisions of the draft questionnaire.

To further test the effectiveness of the questionnaire in actual application, a pre-survey was conducted within the hospital using the revised questionnaire. This helped identify and eliminate potential misunderstandings caused by ambiguous wording in the questionnaire during the actual survey process. The survey was then finalized and distributed.

Participants agreed to the current online survey and clicked the “Continue” button after the consent language in the study design. A convenience sampling approach was utilized for participant selection, primarily due to feasibility considerations regarding participant accessibility and time constraints within the study hospital setting. The inclusion criteria for physicians were that they worked in hospitals, including physicians who had participated in PST services previously and those physicians who were PST services naïve. There were three inclusion criteria for patients: (1) they were first-time visitors to our outpatient clinic; (2) they were conscious and did not suffer from mental illness; and (3) they agreed to complete the electronic questionnaire. The exclusion criteria were as follows: (1) a critical condition and in urgent need of medical treatment; (2) and/or a presence of audio-visual impairment. In total, 120 questionnaires were collected from eligible physicians, of which 113 questionnaires were valid after the removal of 7 incomplete questionnaires, providing a validity rate of 94.2%. In total, 211 questionnaires were collected from patients, of which 206 questionnaires were valid after the removal of 5 incomplete questionnaires, resulting in a validity rate of 97.6%. Before performing the data analysis, we tested the reliability and validity of questionnaires [16,17]. Cronbach’s α was 0.914 for the physician questionnaires and 0.963 for patient questionnaires, deeming a satisfactory result for the internal consistency of instruments. Exploratory factor analysis in structural validity was used to support the validity of the questionnaire. The Kaiser–Meyer–Olkin (KMO) statistics were calculated as 0.821 and 0.914, which passed Bartlett’s test of sphericity (physician: χ^2^ = 1332.40, *p* < 0.001; patient: χ^2^ = 9844.11, *p* < 0.001), indicating that this data was well suited for factor analysis.

### 2.2. Main Variables

The outcome variables of this study (shown in Table 1) were physician and patient dissatisfaction with PST, described by continuous and dichotomous variables, in accordance with previous studies [18,19]. The dissatisfaction evaluation questionnaire collected from the physicians’ perspective contained 20 questions, and the patients’ perspective contained 13 questions. The answers to each question were categorized as not experienced (0), very satisfied (1), satisfied (2), dissatisfied (3), and very dissatisfied (4).

The answers to all the physician perspective questions were summed to obtain the total score of physicians’ dissatisfaction as a continuous variable ranging from 0 to 80, where the higher the score, the higher the level of dissatisfaction. In addition, the degree of dissatisfaction from the physicians’ perspective was classified as a dichotomous variable, with “dissatisfied” and “very dissatisfied” grouped as “dissatisfied” (1), and all others as “other” (0).

The answers to all the patient perspective questions were summed up to obtain the total score of the patients’ dissatisfaction evaluation as a continuous variable ranging from 0 to 52, where the higher the score, the higher the level of dissatisfaction. In addition, the dissatisfaction of the patients’ perspective was classified as a dichotomous variable, with “dissatisfied” and “very dissatisfied” grouped as “dissatisfied” (1), and all others as “other” (0).

### 2.3. Sample Size Calculation

A cross-sectional survey design was used in this study, and sample estimation was based on the utilization of dental cleaning and hygiene services in the population. The sample size was calculated using the following formula [20]:$$n=\frac{\mu_{\alpha /2}^{2}P(1-P)}{\delta^{2}}$$
where N is the sample size; $\mu_{\alpha /2}^{2}$ refers to the statistic of 1.96 for the two-sided test when the confidence interval is 95%; δ is the permissible error at the time of the survey, which was 3% in this study; and P is the utilization rate of dental hygiene services for a given study population. According to the Fourth National Dental Health Epidemiology Survey Report, the utilization rate of dental cleaning services for the whole population in China was 2.2% [21], and the sample size was calculated as follows:$$n=\frac{1.96^{2}\times 0.022(1-0.022)}{0.03^{2}}\approx 91.83$$

A 20% rejection rate was predicted, resulting in a final required sample size of 111 responses. The samples of physicians and patients collected in this survey were 113 and 206, respectively, which met the sample size requirements for the statistical analysis of this study.

### 2.4. Statistical Analysis

Continuous variables were described as ($\bar{x}$ ± s), and categorical variables were described as [*n* (%)]. At the stage of statistical inference, the *t*-test, ANOVA, and $x^{2}$ test was used for univariate analysis and linear regression analysis. Logistic regression analysis was used for multivariate analysis. All data were statistically analyzed using STATA statistical software version 14.0 (StataCorp LP, College Station, TX, USA) and Excel 2016. Statistical significance was considered when *p* < 0.05 (two-sided).

## 3. Results

### 3.1. Demographic Characteristics of Participants

Table 2 presents the demographic and professional characteristics of the participants in the study, which include 113 physicians and 206 patients. In total, 62.83% of physicians had no previous experience with pre-screening triage (PST). Physicians worked in various types of centers, with the highest proportion (34.51%) in dental centers. In total, 13.27% of physicians were senior, and 46.90% of the physicians had worked for more than 7 years. Among patients, 55.83% were female, and the majority of all participants were between 18 and 59 years old. The largest proportion of patient-visited PST centers was the teeth center.

### 3.2. Physician Characteristics

Figure 1 shows the distribution of physician satisfaction ratings of the PST policy, with the horizontal axis representing the percentage of satisfaction and the vertical axis representing the serial number of the question in the corresponding questionnaire. From the physicians’ perspective evaluation, we can see that the item with the highest proportion of “very dissatisfied” is the “willing to experience the PST work for one day and provide suggestions for optimisation”. The item with the highest proportion of “dissatisfied” was “patients will complain about the interface between pre-screening and clinical diagnosis and treatment after pre-screening”. Table 3 shows the distribution of physician dissatisfaction with PST by different characteristics. Physicians with no PST experience had significantly lower dissatisfaction scores (32.51 ± 10.08) compared to those with experience (37.67 ± 9.08; *p* = 0.006). The rate of dissatisfaction was 37.93% for physicians who had PST experience (*p* = 0.071).

Table 4 shows that physicians with PST experience gave scores that were 5.63 points (95% CI: 0.75–10.51) higher than inexperienced physicians when they evaluated the PST services. We analyzed the key factors of physicians’ dissatisfaction with the PST system. Table 4 presents the results of a multifactorial analysis examining the association between various factors and physician dissatisfaction with PST services. Among them, physicians with experience with PST were more likely to be dissatisfied compared to those without experience (OR = 2.42; 95% CI: 0.72–8.15). There were no significant associations found between the type of center, professional title, years of work experience, and physician dissatisfaction (*p* > 0.05).

### 3.3. Patient Characteristics

Figure 1 also presents the distribution of patients’ satisfaction ratings of the PST policy. It can be seen that the patients’ satisfaction with each item is very high. Only the “accompany the patient to the waiting area or consultation room, and the viewing experience is better” had a “very dissatisfied” evaluation, indicating that this part of the participant journey needs to be improved. There was no significant finding on the distribution of patient dissatisfaction with PST by different participant characteristics (Table 3). Table 5 analyzes the key factors of patient dissatisfaction with PST by taking patients as the research object. Patients visiting dental preservation and functional reconstruction center (OR = 1.77; 95% CI: 0.42–7.51) and prosthodontics, implantology, and periodontology center (OR = 1.90; 95% CI: 0.50–7.23) were more likely to be dissatisfied compared to those visiting jaw centers, but were not statistically significant (*p* > 0.05). Patients visiting dental preservation and functional reconstruction centers had higher dissatisfaction than other centers, with nurses’ attitude, smoothness, and experience with patients during the escort service (β = 1.04; 95% CI: 0.07–2.01) being factors that reduce satisfaction.

## 4. Discussion

Pre-screening and triage (PST) implementation provides specific patient information, enabling optimized treatment sequencing and efficient resource use [22]. Studies show strong PST–clinical diagnosis concordance with enhancing service quality [23]. Crucially, PST had improved pandemic safety by screening patients and mitigating infection risks [24,25]. This study integrates PST into the initial diagnostic process of a specialized hospital to reveal the special needs of triage for patients with oral diseases. It provides theoretical support and practical references for constructing a triage system that is more suitable for the service scenarios of dental specialties to improve the efficiency of hospital services.

### 4.1. Decreased Physician Satisfaction Under PST Implementation: Systemic Challenges and the Need for Policy Responses

Analyzed from the perspective of physicians, the providers of healthcare services, our data analysis revealed a counterintuitive finding: Physicians who had experience with PST reported lower satisfaction levels. This finding fundamentally challenges the common assumption that implementing PST can inherently improve physician satisfaction through workflow standardization. While previous studies have positioned PST primarily as a tool for optimizing patient flow [1,2,3], this dissatisfaction highlights the critical systemic and workload issues inherent in PST implementation.

Firstly, PST increases physicians’ workload. While PST streamlines workflows for non-PST physicians, it concentrates additional responsibilities on PST clinicians. These physicians, often fully qualified for independent practice, must not only conduct examinations but also prescribe tests based on symptoms and strategically assign patients to available colleagues [26,27]. This represents a significant expansion of their duties without a proportionate reduction in their existing clinical responsibilities [28,29]. This underscores the critical need for hospital policies addressing equitable workload distribution and role definition when implementing PST. Consequently, PST leads to patients being examined by two physicians. This duplication, while intended for efficiency, can sometimes frustrate patients, potentially straining the physician–patient relationship [30]. Critically, this frustration may escalate into verbal abuse or even physical aggression towards physicians, posing serious workplace safety concerns [31]. This links PST directly to staffing and safety policies, emphasizing the necessity for robust de-escalation training, adequate support staff presence, and clear institutional protocols for managing patient aggression. Finally, PST improves physicians’ ability to diagnose disease through early, comprehensive diagnosis. However, variations in how different physicians communicate, diagnose, and formulate treatment plans can create friction—both between physicians and between physicians and patients [32,33]. This necessitates exceptionally strong inter-departmental communication and coordination mechanisms within the hospital, which may require dedicated administrative support or revised communication protocols, implicating hospital management policies.

In practice, hospitals should take targeted action to increase physician satisfaction with PST work and improve the quality and efficiency of healthcare delivery through developing policies to ensure equitable distribution of workload to prevent burden and role conflicts, and strengthening staffing and safety protocols with an emphasis on patient education on the “two-physician” process rationale. Coupled with de-escalation training and expectation management strategies to improve interdepartmental communication, alongside specialized administrative support, this will ensure a consistent and clear diagnostic process and ensure treatment information is relayed back to the patient and care teams.

### 4.2. Mitigating High Patient Dissatisfaction in Dental Center: A Trinity Approach of Environmental, Process, and Humanistic Enhancements

The phenomenon of high patient dissatisfaction in dental preservation and functional reconstruction centers can be resolved from the dimensions of disease characteristics and service scenarios. Firstly, dental presentations frequently involve diseases that are often accompanied by acute pain, placing patients under significant discomfort, and states of physiological and psychological stress. Empirical evidence consistently shows that pain-induced anxiety readily translates into negative appraisals of the entire service, directly impacting satisfaction metrics. The anxiety triggered by pain is easily transformed into a negative evaluation of the service experience [34,35,36]. The dental preservation and functional reconstruction center of the dental hospital has a high volume of consultations, and the contradiction between the waiting space capacity, diagnosis, and treatment efficiency leads to the prolongation of the patients’ retention. In the closed waiting environment, high-density crowd gathering further aggravates patients, forming a vicious cycle of pain–waiting–dissatisfaction [37,38].

To address this, a holistic strategy is recommended to build a trinity of environment–process–humanity improvement programs. At the level of space optimization, the waiting area can be expanded through the functional reorganization of the consultation area to alleviate the sense of crowding in the physical space. The implementation of an intelligent number calling system and pre-screening terminals enables patients to dynamically grasp the consultation process. In terms of service flow, a green “channel” should be opened for patients with acute endodontists and other emergencies, and an appointment system should be implemented for patients with chronic dental caries to improve the efficiency of consultation through demand stratification. In the dimension of humanistic care, it is proposed to implement a full cycle of emotional management. The waiting area should be set up with an oral health science screen to display videos on pain relief techniques, and nurses should make regular rounds to provide patients with soothing services such as ice packs and cold compresses. The clinic should be equipped with a pain assessment tool, to which the doctor adds a 30 s emotional soothing process in the consultation session to extend pain management from pure medical intervention to psychological soothing.

### 4.3. Limitations and Advantages

This study has several limitations that warrant cautious interpretation. First, this study is a cross-sectional study that relies on a one-time data collection method and therefore cannot determine causal relationships between variables. Instead, it can merely reveal correlations among them. Furthermore, the use of convenience sampling, though pragmatically necessary, given the real-world operational pressures of the participating hospital (e.g., staff shortages preventing randomized recruitment), may have introduced selection bias toward patients with milder conditions or higher literacy abilities. Future multi-center studies with probability sampling are warranted to validate these findings across diverse healthcare settings. Lastly, the timing of this study is notably close to the implementation of the PTS policy. As a result, it may not comprehensively capture the long-term effects of the policy. Future research should incorporate additional validation after the policy has been in operation for a more extended period to gain a more accurate understanding of its long-term consequences.

## 5. Conclusions

This study integrated a pre-screening triage system into the initial consultation process of a specialty hospital, revealing the influencing factors of physician–patient dissatisfaction through data analysis and evaluating the effects of system implementation. This study found that the system was effective in improving triage efficiency and optimizing resource allocation, but there were problems such as insufficient functional adaptation from a physician’s perspective and weak interdepartmental collaboration mechanisms. Challenges include informing the development of targeted improvements for physician workload allocation, new safety protocols, and communication frameworks. This study adds to the research gap in the application of pre-screening triage service in specialized hospitals through empirical data, highlighting the unique triage needs for dental patients, such as specialized initial screening criteria for complex cases, and the mechanism of multidisciplinary collaboration. This study provides theoretical support and a practical reference for constructing a triage system that is more suitable for dental specialty service scenarios through evidence-based workflow adaptations and improving the efficiency of hospital services.

## Figures and Tables

**Figure 1 healthcare-13-01672-f001:**
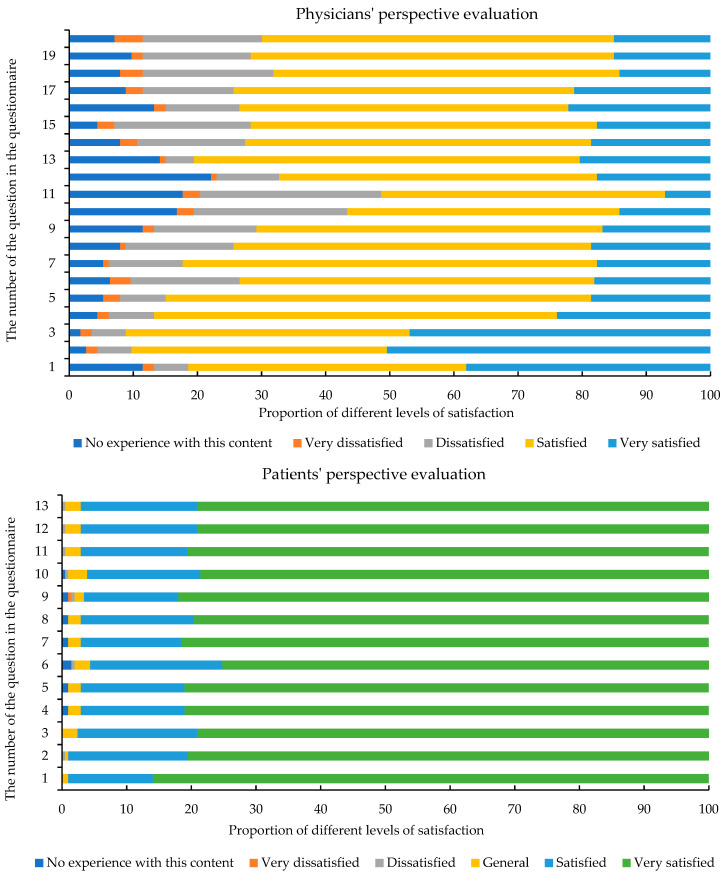
Distribution of physician and patient ratings across dimensions.

**Table 1 healthcare-13-01672-t001:** Division of outcome variables.

Outcome Variables	Classification Methods
Physicians’ Perspective (*n* = 113)	
Continuous variable	The sum of the answers to all questions is the continuous variable for physician dissatisfaction ratings, ranging from 0 to 80, with higher scores being associated with higher levels of dissatisfaction.
Binary variable	The groups dissatisfied and very dissatisfied were combined as dissatisfied and denoted by 1, and the other groups were combined as “Other” and denoted by 0.
Patients’ Perspective (*n* = 206)	
Continuous variable	The sum of the answers to all questions is the continuous variable for physician dissatisfaction ratings, ranging from 0 to 52, with higher scores being associated with higher levels of dissatisfaction.
Binary variable	The groups dissatisfied and very dissatisfied were combined as dissatisfied and denoted by 1, and the other groups were combined as “Other” and denoted by 0.

**Table 2 healthcare-13-01672-t002:** Basic characteristics of participants (*n* = 319).

Variables	Groups	*n*	%
Physicians’ Perspective (*n* = 113)			
Pre-screening triage experience			
	No	42	62.83
	Yes	71	37.17
Type of pre-screening and triage center			
	A	30	26.55
	B	39	34.51
	C	23	20.35
	D	21	18.58
Professional title	Junior	56	49.56
	Intermediate	42	37.17
	Senior	15	13.27
Years of work experience			
	≤3	46	40.71
	4–6	14	12.39
	≥7	53	46.90
Patients’ Perspective (*n* = 206)			
Gender	Male	91	44.17
	Female	115	55.83
Age	≤17 years old	22	10.68
	18–59 years old	151	73.30
	≥60 years old	33	16.02
Type of pre-screening and triage center			
	A	27	13.11
	B	39	18.93
	C	107	51.94
	D	33	16.02

Note: A means dentofacial development management center; B means dental preservation and functional reconstruction center; C means prosthodontics, implantology, and periodontology center; and D means craniomaxillofacial plastic and cosmetic center.

**Table 3 healthcare-13-01672-t003:** Distribution of doctor–patient dissatisfaction with pre-screening triage by different characteristics (*n* = 319).

Variables	Total Score	*p*	Dissatisfaction [*n* (%)]		*p*
	$\left(\bar{x}\pm s\right)$		No	Yes	
Physicians’ Perspective (*n* = 113)					
Pre-screening triage experience		0.006			0.071
No	32.51 ± 10.08		28 (80.00)	7 (20.00)	
Yes	37.67 ± 9.08		36 (62.07)	22 (37.93)	
Type of pre-screening and triage center		0.209			0.152
A	39.86 ± 13.67		7 (50.00)	7 (50.00)	
B	36.51 ± 7.57		22 (62.86)	13 (37.14)	
C	34.17 ± 9.35		18 (78.26)	5 (21.74)	
D	33.38 ± 10.00		17 (80.95)	4 (19.05)	
Professional title		0.512			0.311
Junior	36.67 ± 11.27		29 (63.04)	17 (36.96)	
Intermediate	35.40 ± 7.89		25 (71.43)	10 (28.57)	
Senior	33.08 ± 8.40		10 (83.33)	2 (16.67)	
Years of work experience		0.613			0.448
≤3	35.31 ± 8.14		26 (72.22)	10 (27.78)	
4–6	38.23 ± 16.90		7 (53.85)	6 (46.15)	
≥7	35.34 ± 8.20		31 (70.45)	13 (29.55)	
Patients’ Perspective (*n* = 206)					
Gender		0.315			0.374
Male	15.55 ± 5.53		75 (82.42)	16 (17.58)	
Female	15.90 ± 5.02		89 (77.39)	26 (22.61)	
Age		0.570			0.560
≤17 years old	15.64 ± 5.05		18 (81.82)	4 (18.18)	
18–59 years old	15.57 ± 5.19		122 (80.79)	29 (19.21)	
≥60 years old	16.64 ± 5.63		24 (72.73)	9 (27.27)	
Type of pre-screening and triage center		0.214			0.231
A	14.74 ± 4.04		23 (85.19)	4 (14.81)	
B	16.21 ± 6.17		30 (76.92)	9 (23.08)	
C	16.25 ± 5.43		81 (75.70)	26 (24.30)	
D	14.39 ± 3.98		30 (90.91)	3 (9.09)	

**Table 4 healthcare-13-01672-t004:** Multivariate analysis of physician dissatisfaction with pre-screening and triage.

Variables	Continuous Outcomes						Binary Outcome
	Total Score	P	R	B	D	A	OR (95% CI)
	β (95% CI)	β (95% CI)	β (95% CI)	β (95% CI)	β (95% CI)	β (95% CI)	
Pre-screening triage experience							
No	0	0	0	0	0	0	1.00
Yes	5.63 (0.75, 10.51)	0.45 (−0.39,1.29)	0.37 (−0.64,1.38)	2.35 (0.84,3.85)	1.10 (0.41,1.80)	2.23 (0.14,4.32)	2.42 (0.72,8.15)
Type of pre-screening and triage center							
A	0	0	0	0	0	0	1.00
B	−2.83 (−9.13, 3.48)	−0.95 (−1.90,0.01)	−0.08 (−1.39,1.24)	−0.36 (−2.07,1.35)	−0.15 (−0.94,0.64)	−0.04 (−2.40,2.33)	0.62 (0.16,2.42)
C	−4.21 (−11.15, 2.73)	−0.97 (−2.05,0.10)	−0.28 (−1.72,1.16)	−0.52 (−2.45,1.41)	−0.18 (−1.07,0.71)	−0.35 (−3.03,2.33)	0.36 (0.07,1.72)
D	−5.09 (−11.82, 1.64)	−0.13 (−2.20,0.94)	0.21 (−1.19,1.60)	−1.69 (−3.60,0.22)	−0.29 (−1.17,0.59)	−1.34 (−3.99,1.31)	0.28 (0.06,1.35)
Professional title							
Junior	0	0	0	0	0	0	1.00
Intermediate	−1.51 (−6.31, 3.29)	−0.42 (−1.27,0.43)	−0.52 (−1.52,0.48)	−1.10 (−2.63,0.42)	−0.34 (−1.05,0.36)	−0.28 (−2.39,1.84)	0.62 (0.21,1.86)
Senior	−0.28 (−8.09, 7.53)	0.14 (−1.18,1.47)	−0.80 (−2.46,0.87)	−1.55 (−3.93,0.82)	−0.28 (−1.38,0.81)	1.66 (−1.63,4.95)	0.54 (0.07,4.09)
Years of work experience							
≤3	0	0	0	0	0	0	1.00
4–6	4.36 (−1.98, 10.69)	0.81 (−0.35,1.98)	1.28 (−0.05,2.62)	1.15 (−0.93,3.24)	0.23 (−0.73,1.19)	1.71 (−1.18,4.59)	3.06 (0.71,13.17)
≥7	3.18 (−2.21, 8.56)	0.58 (−0.39,1.55)	0.94 (−0.19,2.07)	1.47 (−0.27,3.21)	0.19 (−0.61,0.99)	0.92 (−1.49,3.33)	2.26 (0.65,7.87)

Note: P means pre-consultation and in-consultation articulation process; R means pre-consultation and in-consultation articulation results; B means changes in patients’ behavior; D means changes in physicians’ behavior; A means physicians’ evaluation.

**Table 5 healthcare-13-01672-t005:** Multifactorial analysis of patient dissatisfaction with pre-screening and triage.

Variables	Continuous Outcomes				Binary Outcome
	Total Score	Physicians	Nurses	Patients	OR (95% CI)
	*β* (95% CI)	*β* (95% CI)	*β* (95% CI)	*β* (95% CI)	
Gender					
Male	0	0	0	0	1.00
Female	0.34 (−1.12,1.80)	0.23 (−0.33,0.80)	−0.12 (−0.62,0.38)	0.23 (−0.30,0.75)	1.37 (0.68,2.78)
Age					
≤17 years old	0	0	0	0	1.00
18–59 years old	−1.01 (−3.68,1.67)	−0.19 (−1.22,0.84)	−0.31 (−1.22,0.61)	−0.51 (−1.47,0.45)	0.74 (0.19,2.85)
≥60 years old	−0.11 (−3.33,3.11)	0.37 (−0.88,1.61)	−0.29 (−1.39,0.81)	−0.19 (−1.34,0.97)	1.11 (0.24,5.20)
Type of pre-screening and triage center					
A	0	0	0	0	1.00
B	1.73 (−1.12,4.57)	0.21 (−0.89,1.31)	1.04 (0.07,2.01)	0.48 (−0.55,1.50)	1.77 (0.42,7.51)
C	1.75 (−0.80,4.31)	0.26 (−0.73,1.25)	0.84 (−0.04,1.71)	0.66 (−0.26,1.58)	1.90 (0.50,7.23)
D	−0.15 (−3.01,2.71)	−0.34 (−1.44,0.77)	0.31 (−0.66,1.29)	−0.13 (−1.16,0.90)	0.58 (0.11,3.13)

Note: Physicians are the pre-screening triage doctors’ attitude towards the patient, the examination of their problems, and the arrangement during the consultation. Nurses mean the nurses’ attitude, smoothness, and experience during the porter service. Patients mean some of the benefits that patients gain through pre-screening triage.

## Data Availability

The raw data supporting the conclusions of this article will be made available by the authors on request.

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
