# Peer review of "Physician and Patient Dissatisfaction with Outpatient Pre-Screening Triage in Public Dental Hospitals: Scope and Strategies for Improvement"

_healthcare, 2025, doi:10.3390/healthcare13141672_

Round 1

Reviewer 1 Report

Comments and Suggestions for Authors

Abstract

  1. The abstract could be more concise and reader-friendly.
  2. Results include too much numerical detail for an abstract without emphasizing interpretation.

Introduction

  1. Initial discussion takes too long to focus on the specific research question.
  2. The differences between PST vs Triage are not clearly and systematically explained.
  3. Citations like [1–3] are overly general and not tied to specific claims.
  4. Several sentences are extremely long and must be divided

Methods

  1. The use of non-random sampling introduces selection bias.
  2. The questionnaire’s validity or reliability is not discussed.
  3. Survey completion vs. drop-out rates are not accounted for.

Results

  1. Dense tables and figure content can be difficult to interpret quickly.
  2. Variables like professional title or age are listed without meaningful discussion.
  3. Terms like “dental center” and “teeth center” are used interchangeably.
  4. Graph axes and legends need improvement for interpretability.

Discussion

  1. Fails to link findings to broader issues like staffing, workload, or hospital policies.
  2. Concepts like “pain-efficiency-satisfaction” need empirical backing.
  3. The findings from one specialty hospital may not be transferable but this is not thoroughly examined.
Comments on the Quality of English Language

Sentences are extremely lengthy

Author Response

Abstract

Comment 1: The abstract could be more concise and reader-friendly.

Response 1: Thank you for pointing this out. We agree with this comment. The abstract has been reorganized to make it more readable and concise, and it's marked in red. The main changes we have made to the abstract are as follows: removal of repetitive method descriptions (e.g., software STATA 14.0 and Excel 2016), consolidation of theoretical associations between conclusions and objectives, use of more precise academic expressions (e.g., “quantified” instead of ‘collected . . . through survey’), highlighting the core values of statistically significant results, and reducing the total word count by 32% from 274 words to 147 words to maintain data integrity and logical chain (Page 1, abstract section, line 12-26).

Comment 2: Results include too much numerical detail for an abstract without emphasizing interpretation.

Response 2: We have rewritten the results section (Page 1, abstract section, line 19-23)

Introduction

Comment 3: Initial discussion takes too long to focus on the specific research question.

Response 3: We have rewritten the first paragraph of the introduction to focus on the definition, role, and differences between prescreening triage and routine triage according to your suggestion (Page 1, introduction section, line 31-39).

Comment 4: The differences between PST vs Triage are not clearly and systematically explained.

Response 4: We have rewritten the distinction between pre-screening triage and routine triage, added new references for support and make sure the new references are appropriate and concise (Page 1, introduction section, line 31-36, references 3 and 4). The modifications are as follows:

“Pre-screening and triage (PST) is one of the effective measures to improve the promotion of high-quality services in public hospitals, it ensures the fair and efficient use of health-care resources through systematic evaluation of medical history, symptom patterns, and vital signs to prioritize critical cases, whereas routine triage conducted by registration clinician/nurses primarily focuses on administrative workflows like department assignment and consultation numbering”.

Comment 5: Citations like [1–3] are overly general and not tied to specific claims.

Response 5: Based on your comments, we have deleted the overly generalized citations [1–3] and re-added new citations that focuses on the definition and role of pre-screening and triage. The specific citations are as follows:

“Mitchell R. Triage for resource-limited emergency care: why it matters. Emergency and Critical Care Medicine, 2023,3(4): 139-141.

Li J, Zhao N, Zhang H, Yang H, Yang J. Roles and Challenges for Village Doctors in COVID-19 Pandemic Prevention and Control in Rural Beijing, China: A Qualitative Study. Front Public Health. 2022 Jun 29;10:888374.

Mitchell R, O'Reilly G, Banks C, Nou G, McKup JJ, Kingston C, Kendino M, Piamnok D, Cameron P. Triage systems in low-resource emergency care settings. Bull World Health Organ. 2025 Mar 1;103(3):204-212.”

Comment 6: Several sentences are extremely long and must be divided

Response 6: Based on your comments, we have split long sentences to optimize language fluency (Page 2, introduction section, line 40-59).

Methods

Comment 7: The use of non-random sampling introduces selection bias.

Response 7: Many thanks for your comment. This study is based on a cross-sectional study using a non-randomized sampling method to obtain data, the study aims to provide a reference to promote the better implementation of pre-screening and triage in hospitals, and further research based on a randomized sampling design is needed. We have added this in the limitations section of this study (Page 10, discussion section, line 298-301). The detailed information is following:

“Furthermore, the convenience sampling technique utilized in this study introduces the potential for selection bias. This sampling strategy may undermine the sample's representativeness, subsequently constraining the generalizability of the findings beyond the specific study setting and the participants involved.”

Comment 8: The questionnaire’s validity or reliability is not discussed.

Response 8: Thanks for your suggestion. The reliability and validity analysis of the questionnaire has been supplemented (Page 3, Method section, line 95-101) and two relevant references have been added as detailed below:

“Before performing the data analysis, we tested the reliability and validity of questionnaire. The Cronbach α was 0.914 for the physician questionnaire and 0.963 for patient questionnaire and internal consistency of instruments was deemed satisfactory. Exploratory factor analysis in structural validity was used to support the validity of the questionnaire. The Kaiser-Meyer-Olkin (KMO) statistics were calculated as 0.821 and 0.914, which passed the Bartlett’s test of sphericity (physician: χ2 = 1332.40, P < 0.001; patient: χ2 = 9844.11, P < 0.001), indicating that this data were well suited for factor analysis.

References:

Cronbach LJ. Coefficient alpha and the internal structure of tests. Psychometrika, 1951, 16, 297–334.

Kimberlin CL, Winterstein AG. Validity and reliability of measurement instruments used in research. Am J Health Syst Pharm. 2008 Dec 1;65(23):2276-84.”

Comment 9: Survey completion vs. drop-out rates are not accounted for.

Response 9: We have added this information in the method section (Page 3, line 91-95). Details are as follows:

“120 questionnaires were collected from the physician's point of view, 113 questionnaires were valid after kicking out 7 incomplete questionnaires, and the validity rate was 94.2%; 211 questionnaires were collected from the patient's point of view, 206 questionnaires were valid after kicking out 5 incomplete questionnaires, and the validity rate of the questionnaires was 97.6%.”

Results

Comments 10: Dense tables and figure content can be difficult to interpret quickly.

Response 10: Thank you very much for your comments. After synthesizing the opinions of the three reviewers, we decided to keep the original number of figures and tables.

Comment 11: Variables like professional title or age are listed without meaningful discussion.

Response 11: Thanks for your comment. The age and gender variables were not statistically supported in the statistical analysis phase of this study and are therefore not described.

Comment 12: Terms like “dental center” and “teeth center” are used interchangeably.

Response 12: Thank you for your suggestion, we have standardized the changes throughout the text, with “dental”.

Comment 13: Graph axes and legends need improvement for interpretability.

Response 13: We have added horizontal and vertical descriptions of Figures 1 to enhance the interpretability according to your suggestion.

Discussion

Comment 14: Fails to link findings to broader issues like staffing, workload, or hospital policies.

Response 14: We have rephrased this paragraph from the physician perspective, we focus on the systemic challenges facing physician workload, patient safety, and hospital management (Page 9, discussion section, line 218-255).

Comment 15: Concepts like “pain-efficiency-satisfaction” need empirical backing.

Response 15: We have deleted similar statements.

Comment 16: The findings from one specialty hospital may not be transferable but this is not thoroughly examined.

Response 16: Thanks for your comment. We have added as one of the study limitations (Page 10, discussion section, line 295-301). Details are as follows:

”This study has several limitations that warrant cautious interpretation. First and foremost, this study is a cross-sectional study that relies on a one-time data collection method and therefore cannot determine causal relationships between variables. Instead, it can merely reveal correlations among them. Furthermore, the convenience sampling technique utilized in this study introduces the potential for selection bias. This sampling strategy may undermine the sample's representativeness, subsequently constraining the generalizability of the findings beyond the specific study setting and the participants involved.”

Reviewer 2 Report

Comments and Suggestions for Authors

The paper remains interesting and it appears that some of the requested revisions have been carried out, however, some adjustments are still needed, as described below:
1. The authors must explain in item 2.1 how the questionnaire questions were prepared, that is, whether they are referenced or created by the authors.
2. The equation presented for calculating the sample size in item 2.3 must be indexed, as defined as the scientific standard for publication.

Author Response

Comment 1: The authors must explain in item 2.1 how the questionnaire questions were prepared, that is, whether they are referenced or created by the authors.

Response 1: Many thanks for your comment. The questionnaire used in this study was designed independently, and we have included the detailed design process in the methods section of the article (Page 2, line 66-82). Details are as follows:

“The survey questionnaire was self-designed. First, during the design phase, based on a literature review, we systematically identified the factors related to doctors' and patients' dissatisfaction with hospitals, while also focusing on the impact factors brought about by the implementation of new policies [10-13]. On this basis, we developed two draft survey questionnaires from the dual perspectives of doctors and patients, focusing on key dimensions such as the triage process, doctor-patient communication, and patients' actual experiences. These questionnaires were scored using a 5-point Likert scale. Second, to ensure the scientific rigor, rationality, and practicality of the questionnaire, a panel of five senior scholars with advanced titles from multiple fields, including hospital management, health policy, and oral medicine, as well as ten healthcare professionals with extensive clinical experience, were organized. Leveraging their professional knowledge and practical experience, they conducted in-depth discussions and meticulous revisions of the questionnaire draft.

To further test the effectiveness of the questionnaire in actual application, a pre-survey was conducted within the hospital using the revised questionnaire. This helped identify and eliminate potential misunderstandings caused by ambiguous wording in the questionnaire during the actual survey process. Ultimately, the questionnaire for this study was finalized.

References:

  • Domagała A, Bała MM, Storman D, Peña-Sánchez JN, Świerz MJ, Kaczmarczyk M, Storman M. Factors Associated with Satisfaction of Hospital Physicians: A Systematic Review on European Data. Int J Environ Res Public Health. 2018 Nov 13;15(11):2546.
  • Deliberato RO, Rocha LL, Lima AH, Santiago CR, Terra JC, Dagan A, Celi LA. Physician satisfaction with a multi-platform digital scheduling system. PLoS One. 2017 Mar 22;12(3):e0174127.
  • Adhikari M, Paudel NR, Mishra SR, Shrestha A, Upadhyaya DP. Patient satisfaction and its socio-demographic correlates in a tertiary public hospital in Nepal: a cross-sectional study. BMC Health Serv Res. 2021 Feb 12;21(1):135.
  • Chen X, Zhang Y, Qin W, Yu Z, Yu J, Lin Y, Li X, Zheng Z, Wang Y. How does overall hospital satisfaction relate to patient experience with nursing care? a cross-sectional study in China. BMJ Open. 2022 Jan 17;12(1):e053899”

Comment 2: The equation presented for calculating the sample size in item 2.3 must be indexed, as defined as the scientific standard for publication.

Response 2: Thank you very much for your suggestion. We have cited authoritative textbooks as reference (reference 18: Yan H, Xu Y. Medical Statistics (3rd Edition) [M]. Beijing: People's Medical Publishing House, 2019.).

Reviewer 3 Report

Comments and Suggestions for Authors

Author Response

Comment 1: Introduction:

The introduction effectively introduces the research topic but needs a clearly stated objective. This lack of a defined objective impacts the discussion section. While the topic introduction is clear, the absence of an explicit research objective weakens the overall impact of the introduction. A well-defined objective would provide a clear roadmap for the reader and ensure that the subsequent sections, particularly the discussion, remain focused and aligned with the study's aims. "The research objective should be explicitly stated, ideally as the final sentence or paragraph before transitioning to the methodology. This placement would provide a clear and concise summary of the study's aim for the reader."

Response 1: Many thanks for your comment. We have rewritten the research objectives section based on your suggestion and placed it separately in the final sentence before the methodology (Page 2, introduction section, line 58-59).

Methods:

The statistical methodology appears sound. However, the description of the data source, the "Evaluation Questionnaire for PST (pre-screening triage) Services in Public Hospitals from the Perspectives of Physicians and Patients," lacks clarity regarding its origin and psychometric properties. The description of the data source needs more detail. It is unclear whether the researchers developed this questionnaire or utilized an existing online survey. Regardless of its origin, it is crucial to explicitly state how the questionnaire's validity (whether it measures what it intends to measure) and reliability (the consistency of its measurements) were established. Providing information on any pilot testing, expert reviews, or statistical analyses conducted to assess these properties is essential for demonstrating the rigor and trustworthiness of the data collected.

The authors should explicitly state:

Comment 2: ▪ "The 'Evaluation Questionnaire for PST Services in Public Hospitals from the Perspectives of Physicians and Patients' was [developed by the researchers for this study/adapted from [cite source if adapted]/an existing validated questionnaire [cite source]]."

Response 2: The questionnaire used in this study was designed independently, and we have included the detailed design process in the methods section of the article (Page 2, line 66-82). Details are as follows:

“The survey questionnaire was self-designed. First, during the design phase, based on a literature review, we systematically identified the factors related to doctors' and patients' dissatisfaction with hospitals, while also focusing on the impact factors brought about by the implementation of new policies [10-13]. On this basis, we developed two draft survey questionnaires from the dual perspectives of doctors and patients, focusing on key dimensions such as the triage process, doctor-patient communication, and patients' actual experiences. These questionnaires were scored using a 5-point Likert scale. Second, to ensure the scientific rigor, rationality, and practicality of the questionnaire, a panel of five senior scholars with advanced titles from multiple fields, including hospital management, health policy, and oral medicine, as well as ten healthcare professionals with extensive clinical experience, were organized. Leveraging their professional knowledge and practical experience, they conducted in-depth discussions and meticulous revisions of the questionnaire draft. Finally, to further test the effectiveness of the questionnaire in actual application, a pre-survey was conducted within the hospital using the revised questionnaire. This helped identify and eliminate potential misunderstandings caused by ambiguous wording in the questionnaire during the actual survey process. Ultimately, the questionnaire for this study was finalized

References:

  • Domagała A, Bała MM, Storman D, Peña-Sánchez JN, Świerz MJ, Kaczmarczyk M, Storman M. Factors Associated with Satisfaction of Hospital Physicians: A Systematic Review on European Data. Int J Environ Res Public Health. 2018 Nov 13;15(11):2546.
  • Deliberato RO, Rocha LL, Lima AH, Santiago CR, Terra JC, Dagan A, Celi LA. Physician satisfaction with a multi-platform digital scheduling system. PLoS One. 2017 Mar 22;12(3):e0174127.
  • Adhikari M, Paudel NR, Mishra SR, Shrestha A, Upadhyaya DP. Patient satisfaction and its socio-demographic correlates in a tertiary public hospital in Nepal: a cross-sectional study. BMC Health Serv Res. 2021 Feb 12;21(1):135.
  • Chen X, Zhang Y, Qin W, Yu Z, Yu J, Lin Y, Li X, Zheng Z, Wang Y. How does overall hospital satisfaction relate to patient experience with nursing care? a cross-sectional study in China. BMJ Open. 2022 Jan 17;12(1):e053899”

Comment 3: ▪ The description jumps directly into data collection without any mention of how the questionnaire's quality was ensured. The authors need to dedicate a substantial portion of this section to detailing the establishment of the questionnaire's validity and reliability.

Response 3: The reliability and validity analysis of the questionnaire has been supplemented (Page 3, Method section, line 95-101) and two relevant references have been added as detailed below:

“Before performing the data analysis, we tested the reliability and validity of questionnaire. The Cronbach α was 0.914 for the physician questionnaire and 0.963 for patient questionnaire and internal consistency of instruments was deemed satisfactory. Exploratory factor analysis in structural validity was used to support the validity of the questionnaire. The Kaiser-Meyer-Olkin (KMO) statistics were calculated as 0.821 and 0.914, which passed the Bartlett’s test of sphericity (physician: χ2 = 1332.40, P < 0.001; patient: χ2 = 9844.11, P < 0.001), indicating that this data were well suited for factor analysis.

References:

Cronbach LJ. Coefficient alpha and the internal structure of tests. Psychometrika, 1951, 16, 297–334.

Kimberlin CL, Winterstein AG. Validity and reliability of measurement instruments used in research. Am J Health Syst Pharm. 2008 Dec 1;65(23):2276-84.”

Comment 4: ▪ Details about Pilot Testing (if conducted): If a pilot study was conducted, the authors should provide details such as: The number of participants in the pilot study (physicians and patients). Any modifications made to the questionnaire based on the pilot study feedback (e.g., changes in wording, item removal).

Response 4: Many thanks for your comment. This information has been added in the method section (Page 2, Method section, line 79-82). The detailed are as following:

“To further test the effectiveness of the questionnaire in actual application, a pre-survey was conducted within the hospital using the revised questionnaire. This helped identify and eliminate potential misunderstandings caused by ambiguous wording in the questionnaire during the actual survey process.”

The specific modifications to the doctor and patient questionnaires have been marked in red in the attached questionnaire. The specific modifications are as follows.

  • We made the following changes to the physician questionnaire:
  • Question P1 was changed from the original question, “Do you agree to click on the triage system during your free time after seeing patients to expect new patients to come for consultation?” to “Do you actively set your status to “available” in the system, hoping to receive patients allocated by PST?”
  • Question A1, “Triage improves the utilization of clinical resources,” has been supplemented with an explanation: “e., the number of patients seeking only consultations has decreased.”
  • The following modification were made to the patient questionnaire:
  • Question 3 has been changed from “The triage doctor can explain the oral problems found during the examination in an organized manner” to “3、The pre-screening triage doctor explained your oral health issues in an organized and understandable manner.”
  • Question 5 has been changed from “The pre-screening triage doctor will assess your oral health issues and arrange for you to see the appropriate department” to “The pre-screening triage doctor assigned you to the most suitable doctor based on your condition.”

Comment 5: Results:

The results section is well-developed but would benefit from objective subheadings to improve clarity and flow. The findings presented are comprehensive. However, organizing them under clear and objective subheadings would significantly enhance the readability and logical flow of this section. This structure would allow readers to easily navigate the key findings related to different aspects of the research question, promoting a smoother transition to the discussion section. Furthermore, considering the data presented in Table 2, potential subheadings for the Results section could include:

  • Demographic Characteristics of Participants: This subheading could introduce the overall sample size and then be further divided (either as sub-points or subsequent paragraphs) to present the demographic breakdown of physicians and patients (e.g., number of participants in each group).
  • Physician Characteristics: This section could detail the specific findings related to the physician sample, potentially using sub-points or paragraphs for:  Pre-screening Triage Experience: Clearly present the percentages and counts of physicians with and without prior PST experience. Type of Pre-screening and Triage Center: Present the distribution of physicians across the different center types (A, B, C, D), explicitly mentioning the center with the highest proportion (dental centers). Professional Titles of Physicians: Clearly state the percentages and counts for junior, intermediate, and senior physicians. Years of Work Experience (Physicians): Present the distribution of physicians' experience duration, highlighting the percentage with more than 7 years of experience.
  • Patient Characteristics: This section would focus on the findings related to the patient sample, potentially using sub-points or paragraphs for:  Gender Distribution of Patients: Clearly state the percentages of male and female patients. Age Distribution of Patients: Present the percentage of patients within the 18-59 age range. Type of Pre-screening and Triage Center Visited by Patients: Explicitly mention the center type with the highest proportion of patient visits (teeth center).

Response 5: Many thanks for your comment. We have reorganized the results section based on your feedback, dividing it into three parts: 3.1 Demographic Characteristics of Participants, 3.2 Physician Characteristics and 3.3 Patient Characteristics. (Page 4-8, Results section, line 152, 162-186, 193-207).

Comment 6: Discussion:

The discussion section currently reads more like a list of recommendations. It should be

rewritten to provide a detailed interpretation of the findings, emphasizing their significance in relation to existing literature and the stated research objective. No new information beyond the presented results should be introduced here. The discussion section needs to shift its focus from making recommendations to providing a thorough analysis and interpretation of the study's findings. This involves highlighting the key takeaways from the results, explaining their implications, and relating them back to the research objective outlined in the introduction (once clarified). It's important to compare and contrast the findings with previous research, acknowledging any similarities or discrepancies and offering potential explanations. Remember, the discussion should build upon the results presented and should not introduce new data or ideas that were not part of the findings. 

For example: While the text states that physicians with PST experience have lower satisfaction, it immediately jumps into reasons why this might be the case. The discussion needs to first interpret what this finding means in the context of the study's objectives. For example, does this finding challenge the assumption that PST improves overall physician satisfaction? Does it highlight unintended consequences of implementing PST? The explanation focuses on the increased workload for pre-screening physicians. While valid, the discussion could be more nuanced. For instance, it could explore which specific aspects of the increased workload are most impactful on satisfaction (e.g., administrative burden, perceived lack of recognition, feeling less connected to the complete patient journey).

Response 6: Many thanks for your comment. Based on your comments, we have revised the discussion section as a whole to address both the physician and patient perspectives separately (Page 9-10, Discussion section, line 218-255). Specific information is provided below:

“Pre-screening and triage (PST) via comprehensive dental examination provides specific patient information, enabling optimized treatment sequencing and efficient resource use [20]. Studies show strong PST-clinical diagnosis concordance, enhancing service quality [21]. Crucially, PST enhanced pandemic safety by screening patients and mitigating infection risks [22,23]. This study integrates the PST into the initial diagnostic process of a specialized hospital to reveal the special needs of triage for patients with oral diseases, and provides theoretical support and practical references for constructing a triage system that is more suitable for the service scenarios of dental specialties and improving the efficiency of hospital services.

Decreased physician satisfaction under PST implementation: systemic challenges and the need for policy responses

Analyzed from the perspective of physicians, the providers of health care services, our data analysis revealed a counterintuitive finding: physicians experienced in PST reported lower satisfaction levels. This findings fundamentally challenge the common assumption that implementing PST can inherently improve physician satisfaction through workflow standardization. While previous studies have positioned PST primarily as a tool for optimizing patient flow [1-3]. This dissatisfaction highlights critical systemic and workload issues inherent in PST implementation. Firstly, PST increased physician's workload. While PST streamlines workflows for non-PST physicians, it concentrates additional responsibilities on PST clinicians. These physicians, often fully qualified for independent practice , must not only conduct examinations but also prescribe tests based on symptoms and strategically assign patients to available colleagues [24,25]. This represents a significant expansion of their duties without a commensurate reduction in other responsibilities, contributing to workload burden and potential role conflict[26,27]. This finding underscores the critical need for hospital policies addressing equitable workload distribution and role definition when implementing PST. Secondly, PST would lead to patients being examined by two physicians. This duplication, while intended for efficiency, can sometimes frustrate patients, potentially straining the physician-patient relationship [28]. Critically, this frustration may escalate into verbal or even physical aggression towards physicians , posing serious workplace safety concerns[29]. This links PST directly to staffing and safety policies, emphasizing the necessity for robust de-escalation training, adequate support staff presence, and clear institutional protocols for managing patient aggression. Finally, PST improves physicians' ability to diagnose disease through early, comprehensive diagnosis. However, variations in how different physicians communicate diagnoses and treatment plans can create friction-both between physicians and between physicians and patients [30,31]. This necessitates exceptionally strong inter-departmental communication and coordination mechanisms within the hospital, which may require dedicated administrative support or revised communication protocols, implicating hospital management policies.”

Comment 7: References:

The reference list appears insufficient, as several statements likely require in-text citations. A thorough review and expansion of the references are necessary, and the inclusion of DOIs is strongly encouraged.

Response 7: Many thanks for your comment. Based on your suggestions, we have reorganized the full text of the literature and added DOI numbers.

Comment 8: Further Action:

A separate section explicitly dedicated to recommendations for physicians regarding PST

(pre-screening triage) services is strongly encouraged. While recommendations might currently be present within the discussion, creating a distinct "Recommendations" section would provide a clear and focused presentation of actionable steps for physicians based directly on the study's findings. This dedicated section would enhance the practical implications of your research and make it easier for readers to identify concrete suggestions for improving PST services.

Response 8: Many thanks for your comment. We have added this targeted action steps in the discussion section (Page 9-10, Discussion section, line 256-261). Specific information is provided below: “In practice, hospitals should take targeted action steps to increase physician satisfaction with PST work and improve the quality and efficiency of healthcare delivery: develop policies to ensure equitable distribution of workload to prevent burden and role conflicts; strengthen staffing and safety protocols and provide de-escalation training; and improve interdepartmental communication through specialized administrative support to ensure consistent and clear diagnostic and treatment information.”

Round 2

Reviewer 1 Report

Comments and Suggestions for Authors

Suggested corrections have been incorporated satisfactorily. 

Comments on the Quality of English Language

Suggested corrections have been incorporated satisfactorily. 

Author Response

Comment 1: Comments and Suggestions for Authors:Suggested corrections have been incorporated satisfactorily.

Response 1: Thank you very much for your valuable comment.

Comment 2: Comments on the Quality of English Language: Suggested corrections have been incorporated satisfactorily..

Response 2: Thank you for your feedback. In order to improve the language quality of the article, we asked a native British speakers to help us polish the language and added them as co-authors (Ravi Lad, Department of Clinical Sciences, Liverpool School of Tropical Medicine). Language changes have been highlighted in red throughout the text.

Reviewer 3 Report

Comments and Suggestions for Authors

Review Report: Physician and patient dissatisfied with outpatient pre-screening triage in public hospitals: Extent and improved strategies

  1. Introduction

The introduction clearly establishes the general importance of PST and its application in emergency and pediatric settings. It effectively highlights the research gap in its application at the "overall hospital level" and, more specifically, within dental healthcare. The study's innovative approach of integrating PST into dental services and collecting "two-way dissatisfaction evaluations" is well articulated, setting a clear objective for the research. The initial paragraphs effectively provide necessary background information and context. My suggestion is to consider slightly broadening the initial problem statement to emphasize the consequences of fragmentation beyond just department-specific optimizations. For instance, how does it affect patient journey or overall hospital efficiency from a holistic perspective?

  1. Methodology

The description of the self-designed questionnaire is comprehensive, including the literature review, expert panel consultation, and pre-survey. This demonstrates a robust approach to instrument development and validation. The reporting of Cronbach's α and Kaiser–Meyer–Olkin (KMO) statistics, along with Bartlett’s test of sphericity, provides good evidence of the questionnaire's reliability and validity. The conclusion that the collected samples met the requirements is clear. The specified statistical tests are appropriate and clearly stated. While convenience sampling is acknowledged as a limitation, explicitly discussing why it was chosen (e.g., feasibility in a specific hospital setting, time constraints) could add context without excusing the limitation.

  1. Results

The results section is well-structured, starting with demographic characteristics and then detailing physician and patient findings separately. Tables and figures are referenced appropriately. Table 2: Provides a clear overview of participant demographics. The note clarifying the PST center types (A, B, C, D) is very helpful. The result section clearly highlights the significant finding regarding physicians with PST experience having higher dissatisfaction scores and the lack of significant associations for other physician characteristics.  The authors should look at (Figure 1), and ensure the percentages on the x-axis sum up to 100% for each question line, reflecting the distribution across all satisfaction levels, or clarify if it's only showing combined dissatisfaction. Also, consider labeling the "Unexperienced" bar more clearly if it represents a specific response option.

  1. Discussion

The discussion effectively interprets the core findings, particularly the "counterintuitive finding" of decreased physician satisfaction under PST, linking it to systemic challenges like increased workload, patient frustration from duplication, and communication variations. This shows a deep understanding of the results.

The discussion consistently references existing literature to support or contrast with the study's findings, strengthening the arguments which is very commendable. In the physician dissatisfaction discussion, when mentioning patient aggression, while safety protocols are suggested, perhaps a brief mention of strategies to manage patient expectations about the "two-physician" process could also be relevant as a preventative measure.

  1. Conclusions

The conclusion is clear, concise, and avoids introducing new information, serving its purpose well. However; the conclusion mentions "insufficient functional adaptation on the physician's side and weak interdepartmental collaboration mechanisms." While these are discussed in the Discussion, a very brief, high-level nod to the solutions proposed in the Discussion (e.g., "informing the development of targeted improvements for physician workflow and interdepartmental communication") could further strengthen the take-home message.

  1. References

There are some references that date back to 1998. I suggest using more recent publications unless the information from those older papers is still highly relevant.

Author Response

Comment 1: Introduction

The introduction clearly establishes the general importance of PST and its application in emergency and pediatric settings. It effectively highlights the research gap in its application at the "overall hospital level" and, more specifically, within dental healthcare. The study's innovative approach of integrating PST into dental services and collecting "two-way dissatisfaction evaluations" is well articulated, setting a clear objective for the research. The initial paragraphs effectively provide necessary background information and context. My suggestion is to consider slightly broadening the initial problem statement to emphasize the consequences of fragmentation beyond just department-specific optimizations. For instance, how does it affect patient journey or overall hospital efficiency from a holistic perspective?

Response 1: Thank you very much for your valuable comment. We have revised the first paragraph of the introduction based on your comments (Page 1-2, introduction section, line 33-47). The modifications are as follows:

“Pre-screening and triage (PST) is a critical mechanism for optimizing healthcare delivery in public hospitals, aiming to balance equitable resource allocation with clinical efficiency [1,2]. While traditional triage conducted by registration staff focuses on administrative tasks—such as department assignment and consultation numbering—this fragmented approach often fails to address systemic inefficiencies in the patient journey [3,4]. Patients may endure repeated assessments across departments resulting in delays in critical care prioritization, or face mismatches between symptom urgency and specialist availability - all of which erode trust in the healthcare system and strain hospital-wide workflows [5,6]. In contrast, PST integrates systematic evaluation of medical history, symptom patterns, and clinical measurements to establish a severity-based prioritization, streamlining the patient pathway [1,2]. PST thereby addresses two interconnected gaps: patient-flow bottlenecks caused by siloed department-level triage, and clinical decision-making delays due to incomplete symptom documentation [7,8]. By synchronizing prioritization with structured symptom documentation, PST not only reduces waiting times for high-risk cases, but also enhances cross-departmental coordination, improving both individual patient outcomes, and operational resilience.”

Comment 2: Methodology

The description of the self-designed questionnaire is comprehensive, including the literature review, expert panel consultation, and pre-survey. This demonstrates a robust approach to instrument development and validation. The reporting of Cronbach's α and Kaiser–Meyer–Olkin (KMO) statistics, along with Bartlett’s test of sphericity, provides good evidence of the questionnaire's reliability and validity. The conclusion that the collected samples met the requirements is clear. The specified statistical tests are appropriate and clearly stated. While convenience sampling is acknowledged as a limitation, explicitly discussing why it was chosen (e.g., feasibility in a specific hospital setting, time constraints) could add context without excusing the limitation.

Response 2: Thank you very much for your valuable comment. We have made revisions to the Methods section and the Limitations section, respectively, (Page 3, Method section, line 94-96; Page 11, Limitation section, line 307-312). The modifications are as follows:

Method section: “A convenience sampling approach was utilized for participant selection, primarily due to feasibility considerations regarding participant accessibility and time constraints within the study hospital setting. ”

Limitation section: “the use of convenience sampling, though pragmatically necessary, given the real-world operational pressures of the participating hospital (e.g., staff shortages preventing randomized recruitment), may have introduced selection bias toward patients with milder conditions or higher literacy abilities. Future multi-center studies with probability sampling are warranted to validate these findings across diverse healthcare settings.”

Comment 3: Results

The results section is well-structured, starting with demographic characteristics and then detailing physician and patient findings separately. Tables and figures are referenced appropriately. Table 2: Provides a clear overview of participant demographics. The note clarifying the PST center types (A, B, C, D) is very helpful. The result section clearly highlights the significant finding regarding physicians with PST experience having higher dissatisfaction scores and the lack of significant associations for other physician characteristics.  The authors should look at (Figure 1), and ensure the percentages on the x-axis sum up to 100% for each question line, reflecting the distribution across all satisfaction levels, or clarify if it's only showing combined dissatisfaction. Also, consider labeling the "Unexperienced" bar more clearly if it represents a specific response option.

Response 3: Thank you very much for your valuable comment. We marked the horizontal axis in Figure 1 with scale lines to ensure that the sum of the percentages for each question row on the x-axis is clearly visible as 100%.

Unexperienced means “No experience with this content”. We have made the corresponding modifications in Figure 1.

Comment 4: Discussion

The discussion effectively interprets the core findings, particularly the "counterintuitive finding" of decreased physician satisfaction under PST, linking it to systemic challenges like increased workload, patient frustration from duplication, and communication variations. This shows a deep understanding of the results. The discussion consistently references existing literature to support or contrast with the study's findings, strengthening the arguments which is very commendable. In the physician dissatisfaction discussion, when mentioning patient aggression, while safety protocols are suggested, perhaps a brief mention of strategies to manage patient expectations about the "two-physician" process could also be relevant as a preventative measure.

Response 4: Thank you very much for your valuable comment. We have revised this section based on your suggestion (Page 10, Discussion section, line 266-270). The modifications are as follows:

“strengthening staffing and safety protocols with an emphasis on patient education on the 'two-physician' process rationale. Coupled with de-escalation training and expectation management strategies to improve interdepartmental communication, alongside specialized administrative support will ensure consistent and clear diagnostic process and ensure treatment information is relayed back to the patient and care teams.”

Comment 5: Conclusions

The conclusion is clear, concise, and avoids introducing new information, serving its purpose well. However; the conclusion mentions "insufficient functional adaptation on the physician's side and weak interdepartmental collaboration mechanisms." While these are discussed in the Discussion, a very brief, high-level nod to the solutions proposed in the Discussion (e.g., "informing the development of targeted improvements for physician workflow and interdepartmental communication") could further strengthen the take-home message.

Response 5: Thank you very much for your valuable comment. We have revised this section based on your suggestion (Page 11, Discussion section, line 323-325, 327-331).

Comment 6: References

There are some references that date back to 1998. I suggest using more recent publications unless the information from those older papers is still highly relevant.

Response 6: Based on your suggestion, we have replaced two older references with more recent ones (Page 12, References section, line 363-368, 397-401, 446-448).